# Molecular Markers in Maternal Blood Exosomes Allow Early Detection of Fetal Alcohol Spectrum Disorders

**DOI:** 10.3390/ijms24010135

**Published:** 2022-12-21

**Authors:** Nune Darbinian, Armine Darbinyan, John Sinard, Gabriel Tatevosian, Nana Merabova, Faith D’Amico, Tarek Khader, Ahsun Bajwa, Diana Martirosyan, Alina K. Gawlinski, Richa Pursnani, Huaqing Zhao, Shohreh Amini, Mary Morrison, Laura Goetzl, Michael E. Selzer

**Affiliations:** 1Center for Neural Repair and Rehabilitation (Shriners Hospitals Pediatric Research Center), Lewis Katz School of Medicine, Temple University, Philadelphia, PA 19140, USA; 2Department of Pathology, Yale University School of Medicine, New Haven, CT 06520, USA; 3Medical College of Wisconsin-Prevea Health, Green Bay, WI 54304, USA; 4Center for Biostatistics and Epidemiology, Department of Biomedical Education and Data Science, Lewis Katz School of Medicine at Temple University, Philadelphia, PA 19140, USA; 5Department of Biology, College of Science and Technology, Temple University, Philadelphia, PA 19122, USA; 6Department of Psychiatry, Lewis Katz School of Medicine at Temple University, Philadelphia, PA 19140, USA; 7Department of Obstetrics & Gynecology, University of Texas, Houston, TX 77030, USA; 8Department of Neurology, Lewis Katz School of Medicine at Temple University, Philadelphia, PA 19140, USA

**Keywords:** alcohol, FAS, fetal development, eye injury, microphthalmia, exosomes

## Abstract

Prenatal alcohol exposure can cause developmental abnormalities (fetal alcohol spectrum disorders; FASD), including small eyes, face and brain, and neurobehavioral deficits. These cannot be detected early in pregnancy with available imaging techniques. Early diagnosis could facilitate development of therapeutic interventions. Banked human fetal brains and eyes at 9–22 weeks’ gestation were paired with maternal blood samples, analyzed for morphometry, protein, and RNA expression, and apoptotic signaling. Alcohol (EtOH)-exposed (maternal self-report) fetuses were compared with unexposed controls matched for fetal age, sex, and maternal race. Fetal brain-derived exosomes (FB-E) were isolated from maternal blood and analyzed for protein, RNA, and apoptotic markers. EtOH use by mothers, assessed by self-report, was associated with reduced fetal eye diameter, brain size, and markers of synaptogenesis. Brain caspase-3 activity was increased. The reduction in eye and brain sizes were highly correlated with amount of EtOH intake and caspase-3 activity. Levels of several biomarkers in FB-E, most strikingly myelin basic protein (MBP; r > 0.9), correlated highly with morphological abnormalities. Reduction in FB-E MBP levels was highly correlated with EtOH exposure (*p* < 1.0 × 10^−10^). Although the morphological features of FAS appear long before they can be detected by live imaging, FB-E in the mother’s blood may contain markers, particularly MBP, that predict FASD.

## 1. Introduction

Fetal alcohol exposure during pregnancy is the leading environmental cause of congenital cognitive impairment in the US [1,2,3]. The adverse effects of prenatal alcohol exposure constitute a continuum of disabilities called “fetal alcohol spectrum disorders” (FASD) [4]. The term FASD encompasses the most severe form, fetal alcohol syndrome (FAS), as well as partial forms, including partial fetal alcohol syndrome (pFAS), alcohol-related neurodevelopmental disorder (ARND), and alcohol-related birth defects (ARBD) [4]. In pFAS, some but not all the diagnostic features of FAS are present. ARBD represents physical abnormalities due to alcohol exposure, while in ARND, none of the facial or physical features are present yet there is a broad range of neurodevelopmental/neurobehavioral problems. FAS occurs in 0.2 to 7 per 1000 children in the US [5,6,7,8]. However, between 3.1% and 9.9% of children have less severe presentations of FASD, while the highest prevalence rates of FASD (13.5–20.8%) have been documented in South Africa.

Craniofacial abnormalities, growth restriction, deficient brain growth with abnormal morphogenesis, or abnormal neurophysiology, and neurobehavioral impairment are diagnostic criteria for FAS [4,9,10]. The cardinal characteristics include a characteristic pattern of facial anomalies, including short palpebral fissures, smooth philtrum, and thin vermilion border of the upper lip, and evidence of CNS neurodevelopmental abnormalities, which may include decreased brain size and other structural brain abnormalities [4]. Studies of children with prenatal alcohol exposure have reported a high frequency of ocular and periocular structural malformations, including microphthalmia, hypertelorism, blepharophimosis, esotropia, atrophy of the disc, hypotrophy of the optic nerve and tortuosity of the retinal vessels [11,12,13,14,15].

Many animal studies of the effects of ethanol (EtOH) on the developing central nervous system (CNS) have emphasized reduced neurogenesis and induction of neuronal apoptosis [16,17,18,19,20,21,22,23,24]. EtOH-induced alterations of oligodendrocyte (OL) development were reported in the fetal brain of mouse [25], macaque [26,27], and human [28]. Prenatal EtOH exposure is associated with excessive OL apoptosis and/or delayed OL maturation in human fetal brain [28,29]. Alcohol administration to mice at gastrulation stages of embryonic development resulted in craniofacial features characteristic of FAS [30,31,32,33,34,35,36], including microcephaly and microphthalmia [37]. It was shown that microencephaly is strongly associated with EtOH exposure during the brain growth spurt [38], a period characterized by rapid glial cell proliferation and maturation, which suggests a potential effect of EtOH on the proliferation, growth, and maturation of glia. In chick and zebrafish embryos, microphthalmia occurs when prenatal EtOH exposure coincided with retinal neurogenesis [39,40]. Although ocular abnormalities have been considered one of diagnostic features of FAS, and were often detected in animal models, there is limited knowledge about eye development at early stages of FAS in humans, and of the mechanisms underlying alcohol toxicity in the developing eye [41,42]. One limitation in human studies is that most EtOH-exposed fetuses do not develop FAS, and current imaging techniques cannot resolve the structural abnormalities diagnostic of FAS early in gestation. Therefore, there is no way of knowing which at-risk fetuses will be born with FAS, limiting our ability to develop therapeutic interventions. Here, we report early developmental effects of alcohol on eye and brain morphology and show that these changes are strongly correlated with biomarkers that can be assayed in fetal-derived exosomes isolated from maternal blood. Characterization of signaling pathways activated in response to prenatal alcohol exposure associated with specific dysmorphic features could lead to identification of early biomarkers for FAS/D in utero. Because of the nature of the project, involving use of human fetal tissue, this limited study could yield definitive data or control for every variable. However, we present sufficient data to suggest larger clinical studies, in which specific molecular biomarkers can be correlated with postnatal emergence of FAS/D.

## 2. Results

Recently, we published findings that prenatal exposure to EtOH induces neuronal and OL injury in human fetal brain in vivo, and in primary cell cultures derived from human fetuses [28,43]. We now present findings from both human fetal brain and eye tissues.

### 2.1. Prenatal EtOH Exposure Inhibits Eye Development in Human Fetuses

We previously established a tissue bank of alcohol-exposed fetuses to investigate in humans the effects of prenatal alcohol (ethanol) exposure on the fetal brain and eye development, and to confirm our preliminary data from the in vivo animal model of prenatal alcohol exposure. To study in utero effects of alcohol on developing eye and facial biomarkers, we used brain and eye tissues from GA 11–21. Fetal eye development in humans begins as early as the 2nd week of gestation and continues to week 37. A timeline for human fetal development is shown in Table 1.

Eye diameters were measured at GA 11–14 weeks Clinical characteristics of subjects following exposure are shown in Table 2.

Eye and pupil diameters were measured at GA 14–21 weeks, using a digital caliper on wet tissues and on fixed frozen histological sections. The eye and pupil diameters were smaller in the EtOH-exposed fetuses than in unexposed controls, although for 2nd trimester eye diameters measured in wet tissues the difference did not reach statistical significance (*p* = 0.055 for Controls and *p* = 0.060 for EtOH-exposed fetuses, Table 3).

### 2.2. Prenatal Alcohol Exposure Inhibits Expression of the CNS and PNS Myelination Markers Myelin Basic Protein (MBP) and Peripheral Myelin Protein-22 (PMP22), Respectively, and Increases Expression of the Apoptosis Marker Activated Caspase-3, during Eye and Brain Development

FACS analysis for MBP, PMP22, and caspase-3 suggested EtOH-dependent changes in the percentage of cells expressing these markers (Figure 1). The EtOH-exposed group showed large, statistically significant reductions in the % of cells expressing the late stage OL marker MBP in fetal brain (*p* = 0.00003 for 1st trimester and *p* = 0.0001 for 2nd trimester, Figure 1A) and eye (*p* = 0.012 and 0.002, Figure 1B). For the OL marker PMP22, the comparable *p* values were *p* = 0.04 and 0.008 in brain (Figure 1C), and *p* = 0.00000001 and 0.004, (Figure 1D) in eye. In unexposed fetuses, PMP22 was expressed in eyes at higher levels than in brain, but EtOH exposure inhibited PMP22 expression in both tissues. The percentage of cells that were positive for activated caspase-3 increased gradually during normal fetal development, but this increase was much greater in the EtOH-exposed fetuses (*p* = 0.001 and *p* = 0.002 in brain, Figure 1E; *p* = 0.00007 and *p* = 0.0001 in eye, Figure 1F), suggesting an alcohol-dependent neurotoxic effect.

### 2.3. Prenatal EtOH Exposure Induces Neuronal and Astrocytic Injury in Fetal Brain and Eye

To characterize the cell types in which apoptotic signaling is activated, we analyzed coexpression of cell lineage-specific markers with activated caspase-3. The percentage of cells co-expressing the neuronal marker β-tubulin and active caspase-3 was increased in EtOH-exposed cases, compared to unexposed controls (*p* = 0.001 in brain and *p* = 0.01 in eye, Figure 2A). Similarly, in EtOH-exposed fetal brain, apoptosis signaling was increased in cells expressing the astrocytic marker GFAP (*p* = 0.03), as well as in neurons (*p* = 0.0005; Figure 2B). We also confirmed our previous finding [28] that these neurotoxic effects of EtOH were more pronounced in male fetuses (*p* = 0.000004) than in females (*p* = 0.02; Figure 2C).

### 2.4. Prenatal EtOH Exposure Inhibits BDNF mRNA and Protein Expression in Fetal Eye, Brain and Synaptosomes

We analyzed BDNF as a marker for injury because fetal brain injuries and other neurodevelopmental disorders often are accompanied by changes in BDNF expression. Prenatal EtOH exposure was associated with a 1.26-fold down-regulation of BDNF mRNA expression in the brain (*p* = 0.0003) and a 4.22-fold down-regulation in the eye (*p* = 0.0002; Figure 3A). This was accompanied by a similar pattern of BDNF protein expression, as measured by ELISA in brain, synaptosomes and eye tissues (Figure 3B). Reduction of BDNF protein expression by EtOH was confirmed by ELISA in both brain (*p* = 0.048) and synaptosomes (*p* = 0.012) but was most marked in eyes (*p* = 0.01), in which ELISA showed that BDNF levels were reduced from 39 ng/mL to 28 ng/mL.

### 2.5. EtOH Dose, Gender, and Race All Strongly Affect Fetal Eye Development

Eye diameters were measured in histological sections of fetal tissues matched by race (Caucasian vs. African American; Table 4), GA (Figure 4), and total amount of EtOH consumed (Figure 4). Previously we demonstrated that prenatal alcohol exposure has a stronger impact on OL differentiation in males than females [28]. In the present study, prenatal alcohol exposure retarded human fetal eye development in a dose-dependent way (*p* = 0.0001, Figure 4A), and this effect was slightly greater in male fetuses than in females, although the sex-related difference was not statistically significant (Figure 4B). The degree of microphthalmia was paralleled by EtOH dose-dependent reductions in eye and brain BDNF levels (ELISA) and PMP22 expression (FACS), and by dramatic increases in brain (FACS; Figure 4C), and eye caspase-3 activity (FACS; Figure 4D). The delay in eye development, as measured by eye diameter, was greater in Caucasian fetuses during both the first and second trimesters than in African Americans, in whom the effect was seen only in the 1st trimester (Table 4).

### 2.6. Prenatal EtOH Exposure Upregulates β-Catenin in Brain and Eye Tissues

One of the earliest biomarkers for eye development is β-catenin, which protects cells from entering the cell cycle in the developing eye. Inactivation of β-catenin is a prerequisite for retinal development [44]. Thus, β-catenin overexpression at early stages may presage developmental delays. The histological structure of normal (alcohol-nonexposed) fetal eye globes and retinas at 16 weeks GA are shown in Figure 5A,B, respectively. To assay β-catenin levels in fetal brain (Figure 5C) and eye (Figure 5D), sections from 2nd trimester fetuses were immuno-stained for β-catenin. The alcohol-exposed brains and eyes had increased levels of β-catenin compared to non-exposed control tissues. EtOH-induced upregulation of β-catenin mRNA expression in 2nd trimester human fetal brains and eyes was confirmed by ddPCR (Figure 5E). Absolute quantitation of β-catenin by ddPCR showed an EtOH-associated increase in mean mRNA levels from 742 copies/μL in control brains to 2253 copies/μL in EtOH-exposed brains (*p* = 0.02), and 870 copies/μL in control eyes to 3443 copies/μL in EtOH-exposed eyes (*p* = 0.01). Thus, exposure to EtOH increases the early expression of β-catenin in the fetal brain and eye and may be a factor in the developmental delays of these structures in FAS.

### 2.7. Brain-Derived Exosomes Provide Biomarkers to Diagnose Early FASD in Fetuses

Previously, several brain-derived exosome markers were studied in relation to fetal exposure to EtOH [45,46]. In the present study, FB-Es isolated from the blood of 10 mothers who used EtOH were compared with exosomes from fetal age- and gender-matched controls with respect to several molecular markers and the fetuses’ eye diameters. For all 10 EtOH-exposed fetuses and their age-matched controls the magnitude of the differences in eye size between the EtOH-exposed group and their matched controls were graphed against the magnitude of the differences in the marker levels. Several correlations are illustrated in Figure 6. The molecular marker showing the greatest correlation with eye diameter was MBP (R = 0.9128, *p* = 0.00061; Figure 6A). Another marker, BDNF, showed no significant correlation (R = 0.41, *p* = 0.2; Figure 6B). Several synaptic proteins, including synapsin-2 (R = 0.86 *p* < 0.001), synaptophysin (R = 0.61, *p* < 0.004), and synaptopodin (R = 0.47, *p* < 0.04), also demonstrated high correlations (not shown). A correlation between eye size and expression levels of FB-E MBP mRNA did not reach statistical significance (R = 0.52, *p* = 0.1; Figure 6C), FB-E miRNA-9, assessed by ddPCR, correlated highly with eye size (R = 0.7034, *p* = 0.02; Figure 6D).

Because of the very high correlation between fetal eye diameter and FB-E MBP content, we examined the feasibility of using this biomarker as a practical predictor of the development of FAS by determining the degree of overlap in FB-E MBP levels between EtOH-exposed and control fetuses, paired only for GA, but not for sex, race or other variables. This yielded a total of 60 EtOH-exposed and 60 control fetuses and matching maternal blood samples. In humans, MBP levels surge during the second trimester, but low levels of MBP are still detectable even in the first trimester [28,29]. FB-E MBP levels were significantly lower in EtOH-exposed fetuses than in unexposed controls, even during the first trimester, with almost no overlap (Figure 7A). The eye diameters of the same fetuses were consistently smaller in the EtOH-exposed group than in controls (Figure 7B). The ratio of eye diameter to FB-E MBP levels was greater in the EtOH-exposed group, but the ratios for both groups were constant throughout the GA range examined (Figure 7C), i.e., the running average curves for the ratios in the two groups paralleled each other throughout the examined GA range, and the slopes were not significantly different from zero. These data suggest that FB-E MBP levels may prove to be a good surrogate marker for predicting whether a child will be born with FAS, even as early as 9 weeks GA.

## 3. Discussion

The present study is significant in two ways. First, it establishes small eye size as a fairly constant anatomical feature of the effects of fetal alcohol exposure that can be detected even in the first trimester. Indeed, the findings in this study suggest that the prevalence of small eyes in newborns exposed to alcohol in utero is higher than previously suspected, and that there are many cases of mild FASD that go undetected early postnatally but could be detected if eye size were measured. Second, the study identifies molecular biomarkers, particularly MBP, that can be quantified in fetal-derived exosomes isolated non-invasively from maternal blood. These two findings may allow us to conduct large-scale population-based studies to determine whether these molecular markers can predict the emergence of FAS/D.

### 3.1. Reduced Eye Size as a Morphological Indicator of Alcohol-Associated Fetal Pathology

The effects of alcohol exposure on fetal development are complex and often severe. Congenital abnormalities may include: Prenatal and/or postnatal growth retardation; CNS involvement (indications of neurological abnormality, developmental delay, or intellectual impairment); microphthalmia and other facial abnormalities, such as small head circumference, short palpebral fissures, flat, and long upper lip, underdeveloped midface, and flattened nose bridge. Many articles have appeared over the years describing these effects and full range of abnormalities or deficits that constitute the fetal alcohol spectrum disorders (FASD) [4,49,50,51]. In 2015, the literature and data on more than 10,000 at-risk children was evaluated thoroughly, and a well-defined criterion for quantifying prenatal alcohol exposure was proposed to set guidelines for diagnosis of individuals prenatally exposed to alcohol. A slightly edited version of these guidelines is reproduced in Appendix A, which enumerates morphological and neurobehavioral criteria for the diagnosis of FAS, pFAS, ARND, and ARBD [4]. Of these, ocular abnormalities are listed in all but ARND. Unfortunately, the effects of alcohol exposure on early fetal development are difficult to determine in utero, using current imaging technology. Thus, the current study selected an anatomical feature of FAS that could be detected in fetuses from electively terminated pregnancies, and to use that feature as an anatomical hallmark against which putative molecular biomarkers could be correlated. In humans, external ocular defects include smaller eye openings, a significant increase in the distance between eyes (telecanthus), drooping of the eyelids (blepharoptosis), and crossed or lazy eyes (strabismus). In animal models, those facial abnormalities can develop if prenatal EtOH exposure occurs after gastrulation, when the three embryonic germ layers are set, but before neurulation, when the neural tube is formed. In humans, that vulnerable developmental period corresponds to between three and six weeks after fertilization [52,53]. During that time, EtOH can induce damage of the cranial neural crest cells, which are critical for development of facial features. Prenatal EtOH exposure results in a marked decrease in cranial neural crest cell proliferation, survival, impaired migration, and apoptosis. The gestational period spanning the end of the first trimester through the second trimester of fetal development is a specific window of vulnerability, because during this period, neural stem cells (NSCs) produce most of the adult neurons, and EtOH has been shown to influence NSC renewal and maturation.

Although eye abnormalities are reported in over 90% of children with FAS [54,55], usually, this is expressed as small palpebral fissures. Actual microphthalmia, i.e., a small, disorganized optic globe, is not, because its diagnostic usefulness is limited by difficulty in detection on physical examination, due to the presence of confounding factors such as microcephaly and short palpebral fissures. The human eye precursor population begins to differentiate three weeks after fertilization, when the neural plate begins to form. The anterior portion of the neural plate gives rise to neuroectoderm, tissues fated to form the CNS, including the retina of the eye. About four weeks after fertilization, the neuroectoderm begins to interact with the surface ectoderm to create tissues that will later give rise to the lens and cornea of the eye. The mesoderm that surrounds the developing eye begins to interact in this developmental process as early as week five, giving rise to the uvea, the sclera, and eyelids. By interfering with that complex developmental process, exposure of the fetus to EtOH during the critical time results in microphthalmia and optic nerve hypoplasia in individuals with FAS. The present study shows that the eye abnormality extends to actual small optic globe diameter (microphthalmia), which appears early in the development of fetuses whose mothers drank EtOH during pregnancy. Eye diameters were smaller in EtOH-exposed cases at least as early as the second trimester, and even earlier (Table 3 and Table 4, Figure 4A,B), and when a larger number of fetuses were measured, paired with controls only for GA, alcohol-exposed fetuses had smaller eyes than unexposed controls, even as early as the 9th week of gestation (Figure 7B). Thus, an anatomical hallmark of FAS (microphthalmia) is present even in the first trimester, and this has allowed us to use eye diameter to validate the prognostic value of putative biomarkers for FAS, such as FB-E MBP levels (see below).

### 3.2. Potential Molecular Biomarkers

Unfortunately, current imaging methods such as ultrasound and MRI, are not sensitive enough to diagnose microphthalmia and other facial features of FAS early in gestation when it might be helpful to diagnose FAS or to identify fetuses at risk. What has been lacking are non-invasively assayable molecular markers that correlate tightly with the morphological features of FAS, and thus could predict the emergence of FAS at an early stage of gestation when therapeutic interventions might be more effective and feasible. To identify such markers, fetal eye diameter proved particularly useful, because the superficial location of the eyes makes them easy to measure, and their small sizes make them unlikely to be damaged during pregnancy termination. The present study has identified several molecular biomarkers that correlate with microphthalmia and can be detected in FB-E obtained from simple maternal blood samples during early stages of gestation. The markers identified in the present study are likely to be useful in predicting significant developmental abnormalities, even when all the criteria for FASD are not met, since eye size appeared to be affected early in gestation and with even small amounts of alcohol use.

The present study showed an association between early EtOH exposure, GA and increased activation of caspase-3, a marker of apoptosis that can be detected before its histological features are apparent, in the fetal brains (Figure 1A) and eyes (Figure 1B) exposed to EtOH, suggesting that the developmental abnormalities seen in FASD are due to excessive apoptosis (Figure 2 and Figure 4C,D). However, a reduction in the number of cells that differentiate into mature brain cells also was suggested by an increase in the levels of β-catenin in the brains of EtOH-exposed fetuses since β-catenin prevents cells from entering the cell cycle during differentiation [56]. Previously, we showed that EtOH exposure increased apoptotic signaling (activated caspase-3) in both OL progenitors and mature OLs, suggesting that the dysmyelination seen in FASD was due to defects in both the generation of mature OLs and apoptosis of mature OLs [28]. The β-Catenin/Wnt signaling pathway participates in many developmental events, and its under-expression was thought to be responsible for the loss of neural crest cells following EtOH exposure in an avian model of FAS [57]. If beta-catenin slows development by taking the cell out of the cell cycle, this suggests that EtOH delays development, not only by apoptosis but by reducing the number of cells that mature into normal eye cells.

Hypoplasia of the optic nerve is common in individuals with FASD and may be responsible for their reduced visual function [51]. Prenatal ethanol exposure can cause optic nerve hypoplasia, one of the leading causes of developmental blindness in children. Hypoplasia occurs when fewer optic nerve axons are developed and maintained, either as a result of premature apoptosis, or due to a reduction in the number of OLs and subsequent dysmyelination. Several alterations in optic nerve myelin have been reported in animal models of FAS, including permanent reduction in myelin thickness, fewer myelinated axons, aberrant myelin sheaths [58,59,60,61]. Ethanol delays the expression of myelin basic protein (MBP) and the maturation of OLs cultured from PD 1–2 rats exposed prenatally to EtOH [29,62], while postnatal EtOH exposure reduces the levels of MBP in the cerebellum of PD15 rats [63,64]. Here, we show that maternal EtOH use during pregnancy is associated with increased neuronal and OL apoptosis and decreased levels of the myelin-associated proteins MBP and PMP22 in fetal brain and eye. PMP22 is a transmembrane component of myelinated fibers in the PNS [65]. In the CNS, PMP22 mRNA is detected during development of the neural crest [66,67], in the cytoplasm of CNS neurons (mRNA and protein; [68]) and in low concentrations in OLs [69]. Previous studies suggested a role for PMP22 in CNS myelin development, even though PMP22 is scarcely detected in the adult CNS [70,71,72]. Finally, our studies showed that prenatal EtOH exposure inhibits BDNF mRNA and protein expression in fetal eye, brain and synaptosomes. Previous reports documented the expression patterns of BDNF in the developing and adult neuroretina, the regulated response of BDNF to retinal and optic nerve injury, and the effects of altered neurotrophin signaling on retinal development [73,74].

The β-catenin signaling pathway plays an essential role in cell proliferation [75,76] and cell fate determination during eye development in mice [77]. Very low β-catenin levels can sustain cell adhesion, but not Wnt signaling [78]. Lack of β-catenin is lethal before gastrulation, but mice heterozygous for β-catenin (Ctnnb1) develop similarly to wild type. in mice. WNT/β-catenin signaling is essential for eye development by controlling the correct patterning of the ocular tissue, promoting the differentiation of the retinal pigment epithelium, controlling the morphogenesis of the optic cup, and maintaining the dorsal retinal identity. Analysis of EtOH-treated mice revealed nuclear accumulation of β-catenin, increased cytosolic expression of p-GSK3β, and increased expression of several β-catenin targets suggesting that EtOH consumption activates the β-catenin signaling pathway [44,56]. Overexpression of β-catenin can induce apoptosis [79], and β-catenin accumulation can be associated with eye injury. Thus β-catenin overexpression has been proposed as an early biomarker for the facial abnormalities of FAS. The present studies have confirmed that exposure to EtOH early in gestation increases β-catenin levels in human brain and eye tissues (Figure 5C–E).

The above findings help to elucidate the pathogenesis of FASD, but do not offer a non-invasive way to determine which at risk fetus is destined to be born with FAS/D. Abnormalities in maternal serum may not be specific to the fetal condition. However, by analyzing the exosomes in maternal blood, using fetal-specific markers, and subdividing the fetal-derived exosomes further into cell type-specific populations, we have isolated a source of fetal brain-specific biomarkers that can be accessed non-invasively (Figure 6A–D). The contents of these FB-Es have suggested biomarker, specifically MBP, whose levels are significantly lower in the fetuses of women who drank alcohol during pregnancy than in fetuses of unexposed controls, with almost no overlap in levels between the two groups, even when a larger population of pregnancies was studied without controlling for factors other than GA. Although MBP levels normally are low in the first trimester, they are not absent [28,29], and the abnormality in MBP levels was detectable well before the second trimester (Figure 7A). The correlation between changes in FB-E MBP protein levels and changes in eye size were particularly strong, and this was true as early as the 9th week of gestation. Moreover, the relationship between FB-E MBP levels and eye diameter was constant, as seen in Figure 7C, suggesting that future large scale clinical studies might focus on FB-E MBP, which may prove to be a practical surrogate marker for facial features (e.g., eye size) to diagnose incipient FAS/FASD [45,46,80].

There are several caveats and limitations to this study. The large variabilities inherent in human studies, requires that the present study be expanded to include larger sample sizes, more precise specification of ethnic, racial, and socioeconomic groups, and expanded controls for potential contributing factors, such as use of tobacco or other substances of abuse, maternal obesity, and maternal psychiatric conditions. Some of these are currently being investigated in our laboratory. In the present study, mothers who used opiates, cocaine or other substances of abuse were specifically excluded. Tobacco use was not an exclusion criterion, but when pairs of fetuses in which either the alcohol exposed or control mother used tobacco were excluded, this did not significantly alter the high degree of correlation between alcohol exposure and reduced eye diameter. Although we believe that in the patient population covered in the present study, self-reporting of alcohol exposure is quite accurate, and if anything, under-reporting of alcohol use would tend to underestimate the biomarker effects described here, a larger study should validate the self-report method with objective blood, hair, or urine testing. Finally, the present study used tissue bank material that was not prospectively accumulated specifically for the study of FAS. Rather we used banked fetal tissues collected on a much more general basis. A follow-up study based on tissues and procedures chosen specifically for the study of FAS/FASD, and funded specifically for this purpose, could overcome these and other limitations.

## 4. Methods

### 4.1. Clinical Samples

A comparison was performed between 20 EtOH-exposed human fetuses and 20 individually matched unexposed controls, selected from among a total of 153 EtOH-exposed and 71 unexposed control elective pregnancy terminations, in which none of the mothers used illicit drugs or CNS-active medications. The selection of cases and controls was driven by the availability of matching maternal blood samples and intact fetal eye and brain tissues, and the availability of data for matching of sex, ethnicity/race, and gestational age (GA), which were not always available in EtOH-exposed and matched un-exposed control subjects. Consenting mothers were enrolled between 11 and 21weeks GA under a protocol approved by our Institutional Review Board (IRB). For each part of the study, all or an appropriate subset of these fetuses were analyzed, matching each EtOH-exposed fetus with its control, at a minimum, with regard to sex and GA.

Cases (EtOH users; see below) were matched to controls (non-users) by GA and fetal sex (10 males and 10 females), depending on biobank availability. Fetuses were enrolled in the order of their acquisition, until the desired number was obtained. At that point, all assays were performed in duplicate. Sex was determined using commercially available SRY primers (Integrated DNA Technologies, Inc., Coralville, IA) and SuperScript One-Step RT-PCR with Platinum Taq (Life Technologies). Immediately following elective pregnancy termination, surgical tissue samples were collected by trained study coordinators; both fresh and snap frozen samples were obtained and transferred to the laboratory within 40–60 min. Then, aliquots were either used for RNA and protein extraction or stored in liquid nitrogen for future use. Initial histologic staining of brain tissues from the Biobank confirmed that we had collected mostly cerebral cortex (see 2.10. Immunohistochemistry). Use of opiates, cocaine or other substances of abuse was an exclusion criterion, although tobacco use was not. Clinical characteristics of participants are summarized in Table 2. Data from both sexes were combined, unless indicated.

*Assessment of Alcohol Exposure in Pregnancy***:** Maternal EtOH exposure was determined with a face-to-face questionnaire that also included questions regarding many types of drugs/medications used, as well as tobacco exposure [28,45,46,80]. EtOH exposure was defined as current daily use, and samples were matched based on the last incidence of alcohol consumption, as indicated by the survey. Alcohol dose was calculated as the total number of drinks consumed in a week multiplied by the number of weeks of exposure. Alcohol exposure was assessed using a detailed questionnaire based on measures adapted from the NICHD PASS study [81]. EtOH consumption for each week since conception (2 weeks after last menstrual period) was self-reported using visual/photographic guides of different types of drinks to estimate actual EtOH dose. Women admitting to any EtOH use were classified as EtOH exposed. Although significant detail on EtOH use was collected, our four most important exposure variables for subsequent analysis were current use (Y/N), exposure pattern, cumulative dose, and average weekly dose. Current use is defined as EtOH use in the 5 days prior to enrollment (Yes/No). Exposure pattern was categorized as social (use is not daily and is <4 drinks on any one occasion), binge (>4 drinks on any occasion), or heavy (daily or near daily EtOH use). Cumulative dose is a continuous outcome that is defined as the estimated total ounces of pure alcohol consumed since conception. Average weekly dose is a continuous outcome defined as cumulative dose divided by the total number of weeks since conception. Women with a known urinary tract infection or a urinalysis with white blood cells or nitrates were not enrolled. Urine was immediately refrigerated to prevent bacterial growth and false negative testing. Although maternal urine ethyl glucuronide (EtG) testing for alcohol consumption often was collected during pregnancy as part of routine clinical care, EtG testing was not included to verify EtOH exposure in the current study. The EtG analysis uses an initial screening immunoassay with a cut-off of 500 ng/mL, adjusted for urine concentration. Sensitivity for EtOH use within the last 120 h at this cutoff is only 70% with a specificity of 99% [82]. We believe this is less accurate than the self-report assessment in our patient population. HPLC-based blood and urine testing for amphetamines and other drugs was performed previously on many of our subjects, to validate or exclude recent drug use in relation to other studies, and we found concordance between reported use and actual use to be greater than 90% in this patient population. Perhaps women who terminate their pregnancy have fewer barriers to truthful reporting than do women in the general population. Moreover, in the general population, errors in self-reports of drug use generally are in the direction of under-reporting. To the extent that some women who denied alcohol use did drink alcohol during their pregnancy, the biomarker differences between controls and alcohol-exposed groups might be even greater than we report.

*Subject Recruitment:* Women reporting alcohol use (or no alcohol use) since conception were grouped from each of the GA windows: 9–14 weeks (1st Trimester), 15–18 weeks (early 2nd Trimester) and 19–23 weeks (late 2nd Tri). GA was determined by a dating ultrasound performed immediately prior to recruitment; at the GAs planned, ultrasound can accurately determine GA ± 10 days [83].

*Sample Collection and Processing:* Fetal brain and eye tissues were collected from each subject. Samples were divided into four groups: (1) Freshly transported on ice for immediate study; (2) snap frozen in liquid nitrogen for protein analysis; (3) stored in RNAlater Stabilization Solution (Thermo Fisher Scientific) for RNA and DNA analysis; and (4) fixed in 4% PFA for immunocytochemistry. Samples collected and stored by our group using our established protocols have demonstrated high levels of RNA recovery (0.7–2.8 µg/µL) and protein preservation (as measured with Coomassie blue staining), compared to fresh samples treated with protease inhibitor. The collection to snap freeze time was less than 30 min. Specimens have been banked for up to 10 years after the completion of a study, to allow for NIH data and sample sharing.

EtOH-using mothers and matching controls were selected from a participating clinic in New Jersey and represent the population in the southern New Jersey and urban Philadelphia area. We aimed for a final sample of 20 EtOH-exposed and 20 case-matched control fetuses that had the requisite combination of maternal blood samples, maternal race matching, fetal GA and sex matching, and well-preserved fetal anatomical structures. As soon as that number was reached, those 40 pregnancies were incorporated into most of the detailed analyses. For some purposes, we could use larger numbers, if the missing data were not required for the analysis. Thus, after analysis of these 40 pregnancies, the most promising biomarkers were selected for larger, but less detailed controlled assessment, involving 60 alcohol-exposed and 60 control fetuses.

### 4.2. RNA Preparation and Real-Time Quantitative Polymerase Chain Reaction (qRT-PCR)

Human fetal or rat total RNA was isolated using the RNeasy kit (Qiagen, Valencia, CA, USA) with on-column DNA digestion. The RT-PCR reaction was performed with 1 μg total RNA, using the One-Step FAST RT-PCR SYBR Green PCR Master Mix (Qiagen). The StepOnePlus Real-Time PCR system thermo cycler was used (Applied Biosystems, Waltham, MA, USA). PCR conditions were: Activation 95 °C 5 min, PCR 45 cycles: 95 °C 10 s, 60 °C 20 s, 72 °C 30 s, melting curve (95–65 °C), cool to 40 °C 30 s. For relative quantification, the expression levels of genes were normalized to the housekeeping gene β-actin. Results are reported as fold regulation [28,84]; see below: Statistical Analysis for qRT-PCR Studies).

### 4.3. Sex Determination using Human Fetal Genomic DNA

Sex determination was carried out using SuperScript One-Step RT-PCR with Platinum Taq (Life Technologies, Way Carlsbad, CA, USA) and a BioRad C1000 Touch Thermal Cycler. In parallel studies, total cellular genomic DNA was isolated from fetal brain tissue for PCR analysis using the QIAamp DNA isolation kit (Qiagen, Germantown, MD, USA) and primers for SRY gene. Amplification was performed in a GeneAmp PCR System 2400. Products were visualized by gel electrophoresis using 2% agarose gel and GelRed DNA stain. The thermal cycler program used was: 45–55 °C for 15–30 min, 94 °C for 2 min, 55–60 °C for 30 s, 68–72 °C for 1 min, 72 °C for 5–10 min, and 12 °C holding temperature. Sex determination in fetal tissue using RNA: RNA was extracted from fetal tissue and One-Step RT-PCR reaction was performed utilizing SRY primers. All PCR reactions were performed in a thermal cycler (C1000 Touch™, BioRad, Hercules, CA, USA) at 94 °C (2 min), followed by 35 cycles of 94 °C (15 s), 65 °C (20 s) and 72 °C (20 s), with a final extension of the cycle at 72 °C for 10 min. The amplified PCR products were separated on 2.5% agarose gels, GelRed stained and visualized under UV transillumination.

### 4.4. Droplet Digital PCR (ddPCR)

For absolute quantitation of mRNA copies, ddPCR was performed using the QX200 ddPCR system. Fifty ng of human fetal total RNA were used with the 1st Strand cDNA Synthesis Kit (Qiagen). After reverse transcription, the cDNA (300 dilution) aliquots were added to BioRad master mix to conduct ddPCR (EvaGreen ddPCR Supermix, BioRad). The prepared ddPCR master mix for each sample (20-μL aliquots) was used for droplet formation. PCR conditions: Activation 95 °C 5 min, PCR 45 cycles at 95 °C 10 s, 60 °C 20 s, 72 °C 30 s, melting curve (95–65 °C), cool to 40 °C 30 s. The absolute quantity of DNA per sample (copies/µL) was calculated using QuantaSoft Analysis Pro Software (Bio-Rad) to analyze ddPCR data for technical errors (Poisson errors) [28]. The Poisson distribution relates the probability of a given number of events occurring independently in a sample when the average rate of occurrence is known and very low. Accurate Poisson analysis requires optimizing the ratio of the number of positive events (positive droplets) to the total number of independent events (the total number of droplets). A greater total number of droplets results in higher accuracy. With 20,000 droplets, the above ddPCR protocol yields a linear dynamic range of detection between 1 and 100,000 target mRNA copies/µL. The estimated error is negligible compared with other error sources, e.g., pipetting, sample processing, and biological variation. The ddPCR data were exported to Microsoft EXCEL for further statistical analysis.

### 4.5. Primers (IDT Inc.)

β-actin: S: 5′-CTACAATGAGCTGCG TGTGGC-3′,

AS: 5′-CAGGTCCAGACGCAGGATGGC-3′,

BDNF: S: 5′-5′-CAGGGGCATAGACAAAAG-3′, AS: 5′-CTTCCCCTTTTAATGGTC-3′,

β-catenin: human CTNNB1, NM_001098209 (OriGene Technologies, Inc., Rockville, MD, USA):

S: 5′-CACAAGCAGAGTGCTGAAGGTG, AS: 5′-GATTCCTGAGAGTCCAAAGACAG

MBP: Myelin Basic Protein (human), S: 5′- ACTATCTCTTCCTCCCAGCTTAAAAA-3′,

AS: 5′-TCCGACTATAAATCGGCTCACA-3′,

Hs miR-9-1 miScript Primer Assay (Qiagen): targets mature miRNA: hsa-miR-9-5p MIMAT0000441: 5′UCUUUGGUUAUCUAGCUGUAUGA MS0001075

Hs SNORD61-11 miScript Primer Assay MS00033705 (for normalization)

SRY: Forward 5′-CAT GAA CGC ATT CAT CGT GTG GTC-3′; reverse 5′-CTG CGG GAA GCA AAC TGC AAT TCT T-3′.

### 4.6. Flow Cytometry

Fresh human fetal brain tissues were washed with cold phosphate-buffered saline (PBS) cocktail with 0.1% BSA and 1% protease inhibitors (Sigma, Bedford, MA, USA). After dissociation of brain tissue through 70 mM mesh, 10,000 cells were placed onto 96-well plates and incubated with fluorescein isothiocyanate (FITC)-conjugated primary antibody for 1 h. Myelin basic protein (MBP) was used as a late OL differentiation/myelination marker in the developing CNS, and peripheral myelin protein 22 (PMP22) was used as a marker of myelinating Schwann cells in the peripheral nervous system (PNS). Early injury in cell populations was quantified based on the proportion of each cell type subset expressing cleaved Caspase-3. Proportions were quantified using 5000 cells and GUAVA FACS (Fluorescence-Activated Cell Sorting) software [28]. Brain tissues (mostly cerebral cortex, as indicated by cytostaining assays; [28]) were used in apoptosis assays.

### 4.7. ELISA Quantification of Brain, Eye, Synaptosomal and Exosomal Proteins

BDNF, MBP and CD81 (American Research Products-Cusabio) were quantified by human-specific ELISAs according to suppliers’ directions.

*Preparation of protein extracts from brain cells for ELISA.* Human fetal brain cells were washed with cold phosphate-buffered saline (PBS) and solubilized in lysis buffer (50 mM Tris-HCl, pH 7.4, 150 mM NaCl, 0.1% Nonidet P-40 and 1% protease inhibitors cocktail (Sigma). Cell debris was removed by centrifugation for 5 min at 4 °C.

*Synaptosome extract preparation.* Human fetal synaptic vesicles and cytoplasmic extracts were prepared from snap frozen brain tissue using the Syn-Per Synaptic Protein Extraction kit (Reagent #87793, Thermo Scientific, Waltham, MA USA). Synaptic vesicle proteins (1 μg) were used in ELISAs.

### 4.8. Immunohistochemistry

Human fetal brain and eye tissue samples either were fixed in 10% buffered formalin and embedded in paraffin, or were used for fast freezing using Tissue-Tek^®^ O.C.T. (optimal cutting temperature compound) embedding medium to quickly embed fresh tissue specimens for cryostat sectioning at 4 μm. Paraffin-embedded tissue was deparaffinated and subjected to antigen retrieval with citrated buffer heated to 97 °C. Permeabilization Buffer (Enzo Life Sciences, Ann Arbor, MI, USA) was utilized as a dilutant (1:10) to permeabilize cell membranes. A primary anti-b-catenin antibody was applied overnight at room temperature. After being rinsed thoroughly with PBS, samples were incubated with FITC-tagged secondary antibodies for 1 h., and mouse monoclonal anti-caspase-3 antibody overnight. After thorough rinsing with PBS, the tissue was incubated in a Texas red-tagged secondary antibody, and 4 mm sections were examined. Fluorescence images of brain and eye tissues were visualized with a Nikon inverted fluorescence microscope equipped with deconvolution software (SlideBook 4.0.1.34; Intelligent Imaging, Denver, CO, USA). Contrast and brightness were adjusted equally for all images using Adobe Photoshop version 5.5.

### 4.9. Antibodies

Anti-human MBP (cat # AB5864), anti-β-catenin and anti-human PMP22 were purchased from Millipore-Sigma (Bedford, MA, USA). Anti-human PMP22 antibody was purchased from Santa Cruz (Santa Cruz, CA, USA). Alexa Fluor^®^-488 labeled anti-cleaved caspase-3 (cat # IC835G, Clone # 269518) was obtained from Life Technologies Corporation (Eugene, OR, USA).

### 4.10. Isolation of Fetal Brain-Derived Exosomes (FB-Es) or Fetal OL-Derived Exosomes (OL-Es) from Maternal Plasma, and ELISA Quantification of Exosomal Proteins

Human FB-Es were isolated as described previously [45,46]. Two hundred and fifty mL of plasma was incubated with 100 mL of thromboplastin-D (Fisher Scientific, Inc., Hanover Park, IL, USA) and cocktails of protease and phosphatase inhibitors. After centrifugation, supernates were incubated with exosome precipitation solution (EXOQ; System Biosciences, Inc., Mountainview, CA, USA), and the resultant suspensions centrifuged at 1500× *g* for 30 min at 4 °C, and pellets resuspended in 400 mL of distilled water with protease and phosphatase inhibitor cocktail for immunochemical enrichment of exosomes. To isolate exosomes from fetal neural sources, total exosome suspensions were incubated for 90 min at 20 °C with 50 mL of 3% bovine serum albumin (BSA) (Thermo Scientific, Inc., Waltham, MA, USA) containing 2 mg of mouse monoclonal IgG1 antihuman contactin-2/TAG1 antibody (clone 372913, R&D Systems, Inc., Minneapolis, MN, USA), that had been biotinylated (EZLink sulfo-NHS-biotin System, Thermo Scientific, Inc.). Then, 10 mL of Streptavidin-Plus UltraLink resin (Pierce, Thermo Scientific, Inc., Waltham, MA USA) in 40 mL of 3% BSA were added, and the incubation continued for 60 min at 20 °C. After centrifugation at 300× *g* for 10 min at 4 °C and removal of supernates, pellets were resuspended in 75 mL of 0.05 mol/L glycine-HCl (pH 3.0), incubated at 4 °C for 10 min and recentrifuged at 4000× *g* for 10 min at 4 °C. Each supernate was mixed in a new 1.5 mL Eppendorf tube with 5 mL of 1 mol/L Tris-HCl (pH 8.0) and 20 mL of 3% BSA, followed by addition of 0.40 mL of mammalian protein extraction reagent (M-PER; Thermo Scientific, Inc.) containing protease and phosphatase inhibitors, prior to storage at −80 °C. For exosome counts, immunoprecipitated pellets were resuspended in 0.25 mL of 0.05 mol/L glycine-HCl (pH = 3.0) at 4 °C, with pH adjusted to 7.0 with 1 mol/L Tris-HCl (pH 8.6). Exosome suspensions were diluted 1:200 to permit counting in the range of 1–5 × 10^8^/mL, with an NS500 nanoparticle tracking system (NanoSight, Amesbury, U.K.). OL-Es were isolated using MBP as a late OL marker from maternal blood (250 μL), to determine whether their membranes show the OL abnormalities or changes previously identified in brain. EtOH-associated dysmyelination was assessed. FB-Es miR-9 was assayed by ddPCR. OL-E MBP protein levels were quantified by ELISA (normalized to exosome marker CD81). MBP, BDNF, and the tetraspanin exosome marker human CD81 (all from American Research Products, Waltham, MA, USA-Cusabio), were quantified by human-specific ELISAs according to the supplier’s instructions. The mean value for all determinations of CD81 in each assay group was set at 1.00, and the relative values for each sample were used to normalize their recovery.

### 4.11. Statistical Analysis

Statistical analysis was described previously [28]. In brief, analysis was performed using SPSS (IBM Corp., released 2017. IBM SPSS Statistics for Windows, Version 25.0. Armonk, NY, USA). A standard method was used for power analysis [85] to determine a minimum number of subjects required in order to detect a treatment effect (effect of EtOH exposure on eye size) with an alpha value of 0.8 (the probability of a type-I error—finding a difference when a difference does not exist) for a significance level of 0.05 and a beta value (the probability of a type-II error—not detecting a difference when one actually exists; Power = 1-β) of 0.2. The results indicated that we would need a sample size of at least n = 19 subjects per treatment group (EtOH-exposed and unexposed controls (we used n = 20), assuming 50% of the population is exposed to alcohol, and 60% of the FASD-diagnosed population has eye abnormalities [54].

All biochemical and molecular assays were performed in triplicate and the measurements averaged to determine a value for each fetus. In each comparison, the mean values for fetuses in each group were compared using both the Student’s *t*-test with Bonferroni correction for multiple comparisons, where relevant (the correction has been applied for multiple comparisons within a figure panel), and by one way ANOVA with Bonferroni correction. Unless otherwise indicated, *p* values shown in the figures are derived by ANOVA. However, both methods gave similar *p* values, and neither method would have led to loss of significance at *p* < 0.05. ANOVA was used for *p* values in all experiments that involved exosomal studies, (Figure 6 and Figure 7).

Proportions were compared using Fisher’s exact test or Chi-Square test. Statistical significance was defined as *p* < 0.05. Data from ddPCR, which measures absolute quantities of DNA per sample (copies/µL), were processed using QuantaSoft Analysis Pro Software (Bio-Rad) to analyze for technical errors (Poisson errors). Data from ddPCR were exported to Microsoft EXCEL for further statistical analysis.

*Statistical analysis for qRT-PCR studies.* In analyzing gene expression data from qRT-PCR assays, we compared gene expression between comparison populations using the well-established and widely adopted ΔΔCT method originally published [86] and described in [87]. We used statistical software purchased from Qiagen to carry out the calculations, which include Fold Regulation (default view), Fold Change, Average ΔCT, 2^−ΔCT, and *p*-value. For more information, the GeneGlobe Data Analysis Center is a complimentary resource for analyzing real-time PCR. The RT PCR modules transform the threshold number of PCR cycles (CT) to calculated results for gene expression. Fold Change is calculated as the ratio of the relative gene expression between the control group and each test group. Numbers greater than 1 indicate increased gene expression, numbers between 0 and 1 indicate decreased gene expression, and a Fold Change value of 1 indicates no change. ΔCT values (the normalized data), for each gene are calculated by subtracting the normalization factor. In comparing two groups, graphing Fold Change would result in skewed distributions, since if test group values are lower than those of the control group, the values are compressed into the range from 0 to less than 1, whereas when the test group values are greater than controls, the range of Fold Change may be much greater. Therefore, in the present study, data are presented as Fold Regulation, which is the customary default mode. For Fold Change values greater than 1, Fold Regulation and Fold Change values are the same. For Fold Change values less than 1, Fold Regulation is the negative inverse of the Fold Change value. Fold Regulation values less than −1 (or Fold Change less than 1) indicate decreased gene expression, and a Fold Regulation value of 1 indicates no change.

The *p*-value for qPCR analyses was based on Student’s *t*-tests, in which each graphed measurement represents the average 2−ΔCT values (or linearized normalized gene expression levels). Thus, in Figure 1, the *p* values represent the cumulative comparison of measurements between the EtOH-exposed fetuses and their matched controls in the indicated GA range. The *p*-value calculation used was based on parametric, unpaired, two-sample equal variance, two-tailed distribution—a method widely accepted in the scientific literature. Both Groups in each pairwise comparison had to contain at least 3 samples for the software to calculate *p* values for that comparison. Repeated measures ANOVA was used to analyze results of qRT-PCR and flow cytometry.

Calculations for correlation studies were based on Spearman’s correlation on exact Two-Tailed Probabilities critical *p* values for N > 2 ≤ 18, and very accurate critical values for N >18 < 101, estimated using the Edgeworth approximation [47,48]. Any *p* values less than 0.05 after correction for multiple comparisons were considered significant.

### 4.12. Ethics, Human Subjects

Consenting mothers were enrolled between 11 and 21weeks gestation, under a protocol approved by our Institutional Review Board (IRB). This protocol involved no invasive procedures other than routine care. Maternal EtOH exposure was determined with a face-to-face questionnaire that also included questions regarding many types of drugs/medications used, as well as tobacco exposure [28,45,46,80]. The questionnaire was adapted from that designed to identify and quantify maternal EtOH exposure in the NIH/NIAAA Prenatal Alcohol and SIDS and Stillbirth (PASS) study [81].

All procedures involving collection and processing of blood and tissues were done according to NIH Guidelines through a trained Study Coordinator. All investigators were trained annually to complete Citi Program—Human Subject training, Biohazard Waste Safety Training and Blood–Borne Pathogens Training, and all other required training.

a.**Eligibility Criteria:** The blood and tissue samples were obtained according to NIH Guidelines through a trained Study Coordinator. Samples were collected regardless of sex, ethnic background, and race.b.**Treatment Plan:** Each patient was asked to sign a separate consent form for research on blood and tissue samples. Blood obtained was processed for collection of serum and plasma. No invasive procedures were performed on the mother, other than those used in her routine medical care. Fetal tissues were processed for RNA or protein isolation.c.**Risk and Benefits:** There are very small risks of loss of privacy as with any research study where protected health information is viewed. The samples were depersonalized before they were sent to the lab for analysis. There were no additional risks of blood sampling as this was only performed in patients with clinically indicated venous access. There was little anticipated risk from obtaining approximately 2–3 cc of blood, but a well-trained Study Coordinator collected all samples.

There was no direct benefit to the research subjects from participation, but there is significant potential benefit for the future FAS subjects and the general population. This research represents a reasonable opportunity to further the understanding, prevention, or alleviation of a serious problem affecting the health or welfare of FAS patients.

d.**Informed Consent:** Consent forms were maintained by the Study Coordinator and were not sent to the investigator with the samples. The de-identified log sheets and IRB protocol were sent by the Study Coordinator to Principal Investigator with each blood and tissue sample. This sheet contains an assigned accession number, the age, sex, ethnicity, and race of the patient. Except for an assigned accession number, no identification was kept on the blood and tissue samples.

## 5. Conclusions

The number of children diagnosed with FAS is a small proportion of those prenatally exposed to alcohol, but the frequency with which FAS is diagnosed has been increasing. Moreover, exposure to doses lower than those that result in the most severe manifestation of prenatal exposure to alcohol, FAS, still carries risks for non-trivial developmental abnormalities. Moreover, FAS remains underdiagnosed. Recent random testing of schoolchildren by public health facilities resulted in the establishment of newly diagnosed cases of FAS, microcephaly, autism, and cerebral palsy, all associated with prenatal alcohol exposure. Thus, our findings may have implications that go beyond FAS. In clinical practice, the diagnosis of FAS(D) is not made prenatally. Neither ultrasound nor MRI is sensitive enough to discern the morphological features during the first trimester, and there are no molecular biomarkers that predict the emergence of FAS(D) postnatally. The present study demonstrates that the microphthalmia characteristic of FAS is present very early during fetal development, and is highly correlated with molecular markers (e.g., low levels of MBP and PMP22 mRNA and protein, and elevated levels of activated caspase-3) in fetal cell type-specific exosomes. These can be isolated noninvasively from maternal blood samples, making them potentially valuable as a source of biomarkers for FAS(D). The correlation between changes in FB-E MBP protein levels and changes in eye size were particularly strong. Future studies may establish these and other biomarkers as useful surrogates for the facial and brain abnormalities that define FAS. Such studies will determine whether normal biomarker levels can provide reassurance to a mother who drank alcohol before she knew she was pregnant, that her child will not develop FAS(D). If these molecular markers were developed, they could be incorporated into routine blood test analysis during pregnancy, and at early postnatal times. In a 1992 national probability survey of the public, sponsored by the March of Dimes, 38 percent of respondents said that new types of genetic testing should be suspended until the privacy issues are settled (Institute of Medicine (US) Committee on Assessing Genetic Risks, 1994 [88]). Each newly developed genetic test raises serious issues for medicine, public health, and social policy about how the test results should be used. Similar considerations apply to the molecular markers suggested by the present study. Although blood/exosomal assays are not genetic tests, mothers still should be allowed to choose or refuse the biomarker screening. These implications and the potential benefits of such blood/exosomal test should be thoroughly explained to the mothers, who should be protected from unethical treatment by third parties who may seek the information to unethically violate the mother’s autonomy, confidentiality, privacy, and equity. Any clinical trial also should include such protections. Finally, the biomarkers could have mechanistic relevance, and thus inform research aimed at devising therapeutic interventions, e.g., antiapoptotic drugs, or molecular approaches to enhance MBP transcription, by targeting the affected biomarkers, or their promotors and suppressors.

## Figures and Tables

**Figure 1 ijms-24-00135-f001:**
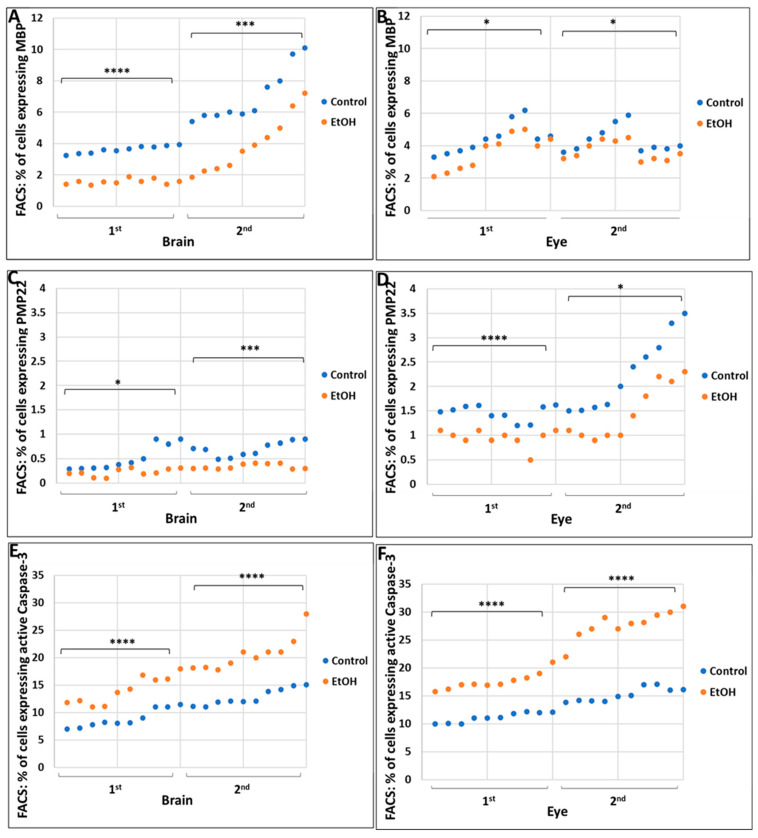
Prenatal alcohol exposure inhibits expression of markers for mature myelin, MBP and PMP22, and activates caspase-3 in human fetal brain and eye. The percentage of cells expressing molecular markers were determined by FACS in brains and eyes of fetuses exposed to EtOH, and compared with non-exposed controls. (**A**) EtOH-dependent reduction in the proportion of cells expressing the late stage OL marker, MBP, in brain and eye (**B**). (**C**) EtOH-dependent reduction of the expression of Schwann cell marker PMP22 in brain and eye (**D**). (**E**) EtOH-induced increases in the proportion of cells expressing the pro-apoptotic marker, activated caspase-3 in brain and eye (**F**). Twenty EtOH-exposed fetuses were compared with 20 individually matched unexposed controls, 10 matched pairs from the 1st trimester and 10 from the 2nd trimester. Each dot represents the average of triplicate measurements for that fetus. Significance levels are for the comparison between all EtOH-exposed and all unexposed controls within the indicated trimester (1st or 2nd), based on ANOVA. * *p* < 0.05; *** *p* < 0.001; **** *p* < 0.0001.

**Figure 2 ijms-24-00135-f002:**
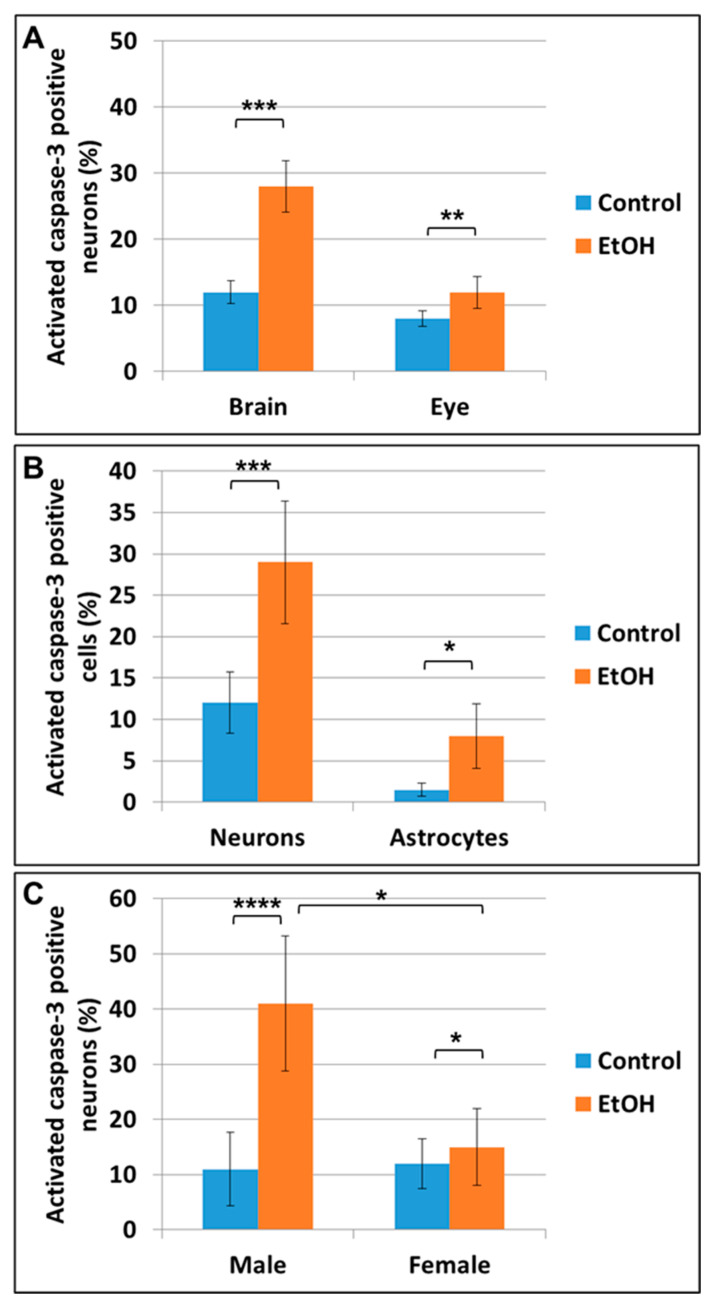
Prenatal EtOH exposure induces neural injury in fetal brain and eye, and affects male fetuses more than females. Human fetal brain and eye tissues from 20 EtOH-exposed fetuses and 20 matched unexposed controls were dissociated through 70 mM mesh, and cells were assayed by FACS using β-Tubulin as a neuronal marker, GFAP as a glial marker, and activated caspase-3 as an apoptotic marker. The value for each fetus was the average of the triplicate determinations. The % of each cell type that was labeled for activated caspase-3 was determined and evaluated with respect to fetal sex and in vivo exposure to EtOH. (**A**) In utero exposure to EtOH increased apoptotic signaling in neurons of fetal brains and eyes. β-Tubulin was used as a neuronal marker. (**B**) Apoptotic signaling was increased in astrocytes and neurons in EtOH-exposed brains (*n* = 20 fetuses). (**C**) The effect of exposure to EtOH on neuronal apoptotic signaling was greater in male fetuses (3.65-fold) than in females (1.50-fold). Each assay was performed in triplicate. Error bars indicate standard deviations based on numbers of fetuses, not the number of triplicated assays. * *p* < 0.05; ** *p* < 0.01; *** *p* < 0.001; **** *p* < 0.0001. In (**C**), the *p* value * above the top line linking the male and female groups refers to the difference between male and female fetuses regarding the fold changes in apoptotic signaling in EtOH-exposed vs. non-exposed control fetuses.

**Figure 3 ijms-24-00135-f003:**
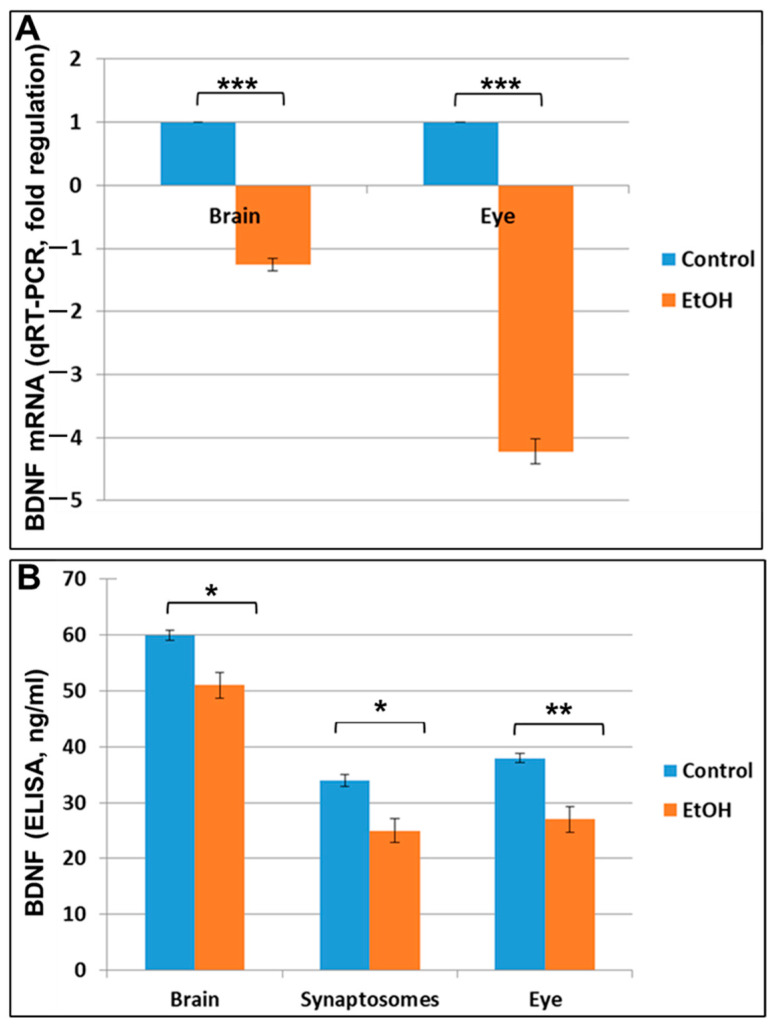
Prenatal EtOH exposure inhibits BDNF mRNA and protein expression in fetal eye, brain and synaptosomes in EtOH-exposed samples compared to unexposed controls. (**A**) Prenatal EtOH exposure was associated with a 1.26-fold down-regulation of BDNF mRNA expression in brain, and a 4.22-fold down-regulation in eye (Control *n* = 10 and EtOH *n* = 10, each representing the average of triplicate determinations). There was a 2-fold decrease in BDNF mRNA expression in eye tissues compared to brain tissues. (**B**) Expression of BDNF protein in brain, synaptosomes and eye was assayed by ELISA (see Methods). Each assay was performed in triplicate and error bars indicate standard deviation. Significance levels are based on the number of fetuses, not the number of measurements. * *p* < 0.05; ** *p* < 0.01, *** *p* < 0.001.

**Figure 4 ijms-24-00135-f004:**
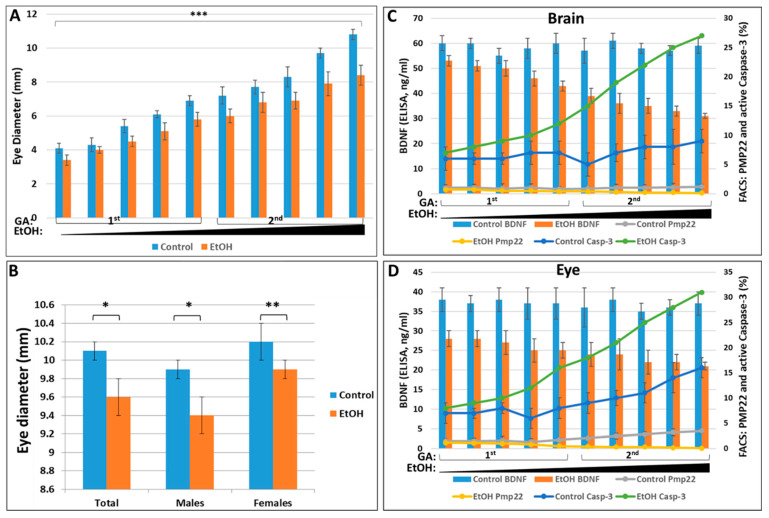
Influence of sex and EtOH dose on the effects of in utero exposure to alcohol on human fetal eye development during 1st and 2nd trimesters in EtOH-exposed samples compared to unexposed controls. (**A**) Direct correlation between reduction in eye diameter and total alcohol dose. Samples from 10 pairs of fetuses GA- and sex-matched with fetuses whose mothers’ last drink (if any) occurred before pregnancy, arrayed within each trimester by the estimated total EtOH exposure. The estimated alcohol dose was calculated as the product of the total drinks in a week, the alcohol dose per drink, and the number of weeks of exposure. The *p*-value *** refers to the mean eye diameter for all EtOH-exposed fetuses vs. the means for the GA- and sex-matched controls. (**B**) In the same 10 matched pairs of fetuses, exposure to EtOH was associated with reduced eye diameters. Fetal sex was determined by RT-PCR with SRY primers. Eye diameter was measured on histological sections. The *p*-values between EtOH-exposed and unexposed controls are based on the number of fetuses studied, not the number of triplicate measurements. Although the fold change associated with exposure to EtOH was slightly greater in males (0.04-fold reduction in eye diameter) than in females (0.03-fold reduction), this was not statistically significant (*p* = 0.09). * *p* < 0.05; ** *p* < 0.01. Reduction in BDNF and PMP22 levels was correlated with caspase-3 activation, and with alcohol dose during 1st and 2nd trimesters in brain (**C**), and in fetal eye (**D**). Correlation between alcohol dose and expression of PMP22, BDNF and active caspase-3 in brain and eye (BDNF is presented as bars on the primary y-axis (left); PMP22 and active caspase-3 are presented as lines on the secondary y-axis (right). Total cumulative alcohol dose for 1st trimester was 57–168 (or 12–30 drinks/month), and for 2nd trimester was from 54.4 to 1827.5 (or 6–320 drinks/month). Each assay was performed in triplicate and in frames (**A**,**C**,**D**), each bar represents one fetus, and the variability of the measurement technique is indicated by standard deviations based on those three measurements.

**Figure 5 ijms-24-00135-f005:**
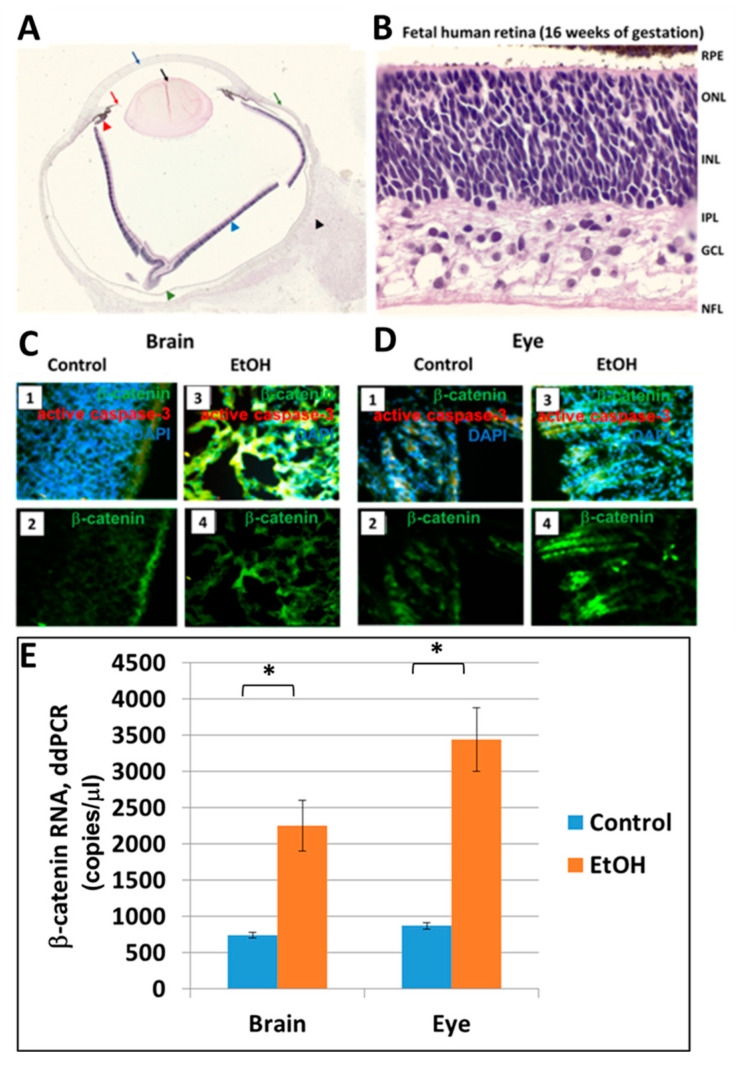
Prenatal EtOH exposure upregulates β-catenin in fetal brain and eye in EtOH-exposed fetuses compared to matched unexposed controls. (**A**) Fetal eye globe (16 weeks GA) in sagittal section: the cornea (cyan arrow), iris (red arrow), lens (black solid arrow), and ciliary body (red arrowhead) can be identified. The anterior chamber is the space between the anterior iris and posterior cornea, and the posterior chamber is comprised of the space between the vitreous and the posterior aspect of the iris. In the field of the posterior globe, the head of the optic nerve (black arrowhead), the retinal pigment epithelium (RPE) (green arrowhead), retina (cyan arrowhead), and the sclera (green arrow) can be identified. (**B**) Fetal human retina (16 weeks GA). Hematoxylin and eosin staining. RPE = retinal pigment epithelium; ONL = outer nuclear layer, containing the nuclei of the future photoreceptors (the photoreceptor outer segments do not yet exist); INL = inner nuclear layer, containing nuclei of the Müller cells, and the where the horizontal, bipolar and amacrine cell bodies will be; IPA = inner plexiform layer; GCL = ganglion cell layer; NFL = nerve fibre layer. (**C**) Effect of EtOH exposure on β-catenin in fetal brain. Brain sections were collected and prepared from EtOH-exposed 2nd trimester fetuses and unexposed 2nd trimester controls, and stained with DAPI (blue) and caspase-3 (red) (panels 1 and 3). Panels 2 and 4 show staining for β-catenin only (green). (**D**) Effect of EtOH exposure on β-catenin in fetal eye. Panels 1 and 3: β-catenin and DAPI staining. Brain and eye sections were prepared from EtOH-exposed 2nd trimester fetuses and stained with DAPI (blue, which appears as cyan because of overlap with green) and activated caspase-3 (red which appears as yellow because of overlap with green, or as magenta because of overlap with blue, or as light yellow because of overlap with blue and green). Panels 2 and 4, β-catenin (green). (**E**) EtOH-induced upregulation of β-catenin mRNA expression in 2nd trimester human fetal brain and eye. Absolute quantitation of β-catenin mRNA (copies/μL) was obtained by ddPCR for 10 EtOH-exposed and 10 age- and sex-matched control brains and eyes. Each assay was performed in triplicate and averaged for each fetus. Error bars are SD among the averaged values for the fetuses. * *p* < 0.05 (*p* = 0.02).

**Figure 6 ijms-24-00135-f006:**
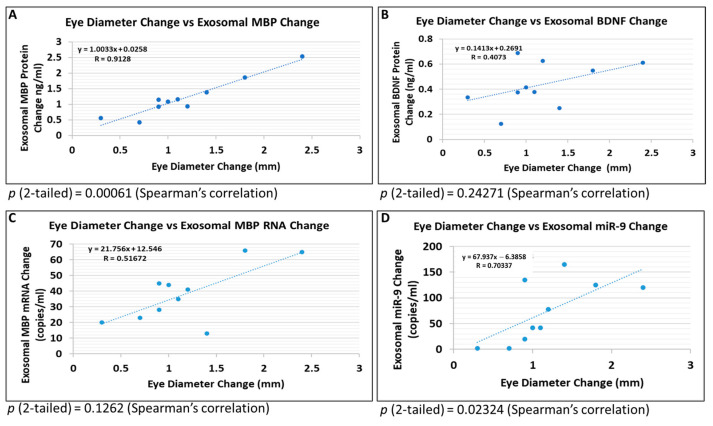
Correlations between human fetal eye size and fetal brain-derived exosomal molecular markers. Eye diameters were measured in histological sections from EtOH-exposed samples and unexposed controls. Protein levels were measured by ELISA (for MBP and BDNF). RNA copy numbers were measured by ddPCR (miR-9 and MBP). Ten EtOH-exposed fetal eye tissues each were paired with an age- and sex-matched control, and with their matching maternal blood samples from 1st and 2nd trimester pregnancies. Assays were performed in triplicate on contents of FB-Es isolated from the maternal blood. Each dot represents the ratio of the values for each fetus vs. its matched control. (**A**) Correlation between change in eye size (difference between EtOH-exposed fetus and its paired control) and changes in exosomal MBP protein levels presented as a scatter plot. (**B**) EtOH-associated changes in exosomal BDNF protein levels (*p* = 0.1262, not significant). (**C**) Correlation between eye diameter changes and changes in FB-E miR-9 copy numbers (*p* = 0.24271, not significant). (**D**) As in (**C**), but for OL-E MBP mRNA copy numbers. Calculations are based on Spearman’s correlation on exact Two-Tailed Probabilities critical *p* values for N > 2 ≤ 18, and very accurate critical values for N > 18 < 101, estimated using the Edgeworth approximation [47,48]. Only *p*-values less than 0.05 (**A**,**D**) are considered significant.

**Figure 7 ijms-24-00135-f007:**
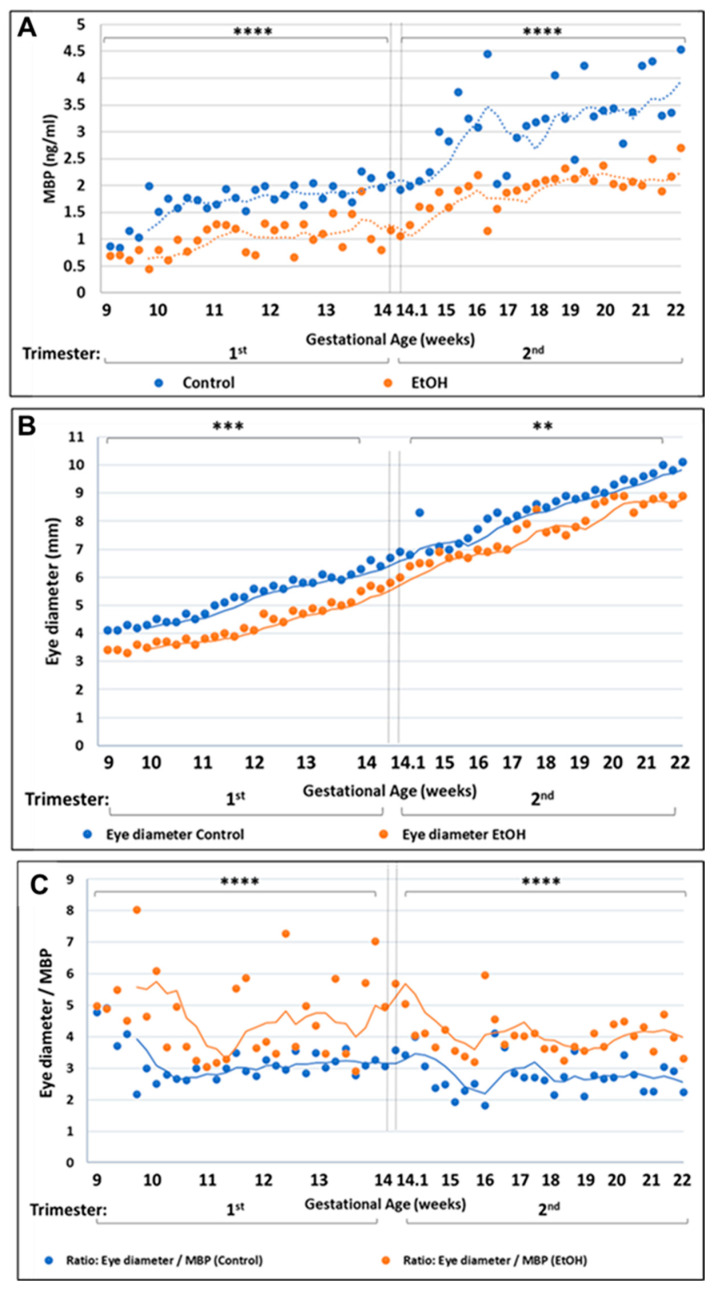
MBP levels in fetal brain-derived exosomes, and eye diameter changes during development in EtOH-exposed samples and unexposed controls. FB-Es were isolated from the blood of 60 mothers who drank alcohol during pregnancy and 60 GA-matched unexposed controls. (**A**) MBP protein levels in the FB-Es were measured by ELISA. Unlike the pregnancies used in the previous figures, the sample is much larger, and matching involved no other variables (fetal sex, maternal age, maternal race, etc.), so that we could assess how generally useful this molecular marker might be for identifying those at-risk fetuses that were destined to be born with FAS. First trimester pregnancies ranged from 9 to 14 weeks GA, and the 2nd trimester pregnancies ranged from 14.1 to 22 weeks GA. The small dotted lines track the moving averages of 5 data points. Note that the control levels of MBP rose dramatically early in the second trimester, whereas the MBP levels of the EtOH-exposed group did not show this sudden jump. In both the first and second trimesters, the differences between the EtOH and control groups were highly significant, with *p* values for the comparison of fetal exosomal MBP levels of all first trimester and second trimester EtOH-exposed fetuses with those of their age-matched controls much lower than 0.00001. **** *p* < 0.0001, comparing EtOH-exposed fetuses with unexposed controls. (**B**) Eye diameter changes in 60 fetuses. Each assay was performed in triplicate and the values indicated by the dots are the averages of the three determinations. ** *p* < 0.01; *** *p* < 0.001. Sixty EtOH-exposed cases were compared with 60 controls during 1st trimester and 2nd trimester. Eye diameters were measured in histological sections. (**C**) Ratio between eye diameters (mm) and MBP levels (ng) in 60 EtOH-exposed fetuses and 60 control fetuses. **** *p* < 0.00001 (*p* = 4.55912 × 10^−6^ and *p* = 2.61042 × 10^−10^ in 1st and 2nd trimester, respectively).

**Table 1 ijms-24-00135-t001:** Timeline of human fetal organ development.

	Gestation Period in Weeks	1	4	7	10	13	16	19	22	28	31	34	37
Alcohol	
Brain	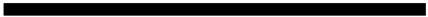
Heart	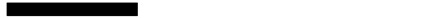
Arms	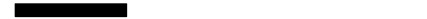
Eyes	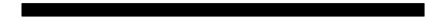
Legs	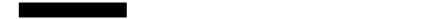
Teeth	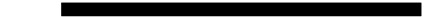
Palate	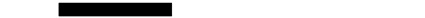
Ears	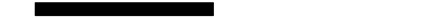
Genital Area	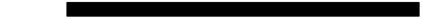

**Table 2 ijms-24-00135-t002:** Clinical characteristics of pregnant subjects used in most of the experiments.

	Controls (n = 20)	EtOH Users (n = 20)
Maternal age (years ±SD)	24.15 ± 2.3	28.0 ± 2.7
Parity (±SD)	0.83 ± 0.42	0.82 ± 0.67
GA (weeks ±SD)	14.92 ± 1.58	15.22 ± 1.6
GA range (weeks)	11.1 to 21	11.5 to 21
Body Mass Index (±SD)	24.4 ± 3.3	26.1 ± 1.5
Hispanic Ethnicity (%)	15	15
Race		
White (%)	50	50
Black (%)	50	50
Tobacco Use (%)	20	25
Male Fetal Sex (%)	50	50

**Table 3 ijms-24-00135-t003:** Prenatal alcohol exposure inhibits fetal eye development in human fetuses (GA 14-21 weeks, 2nd trimester): Comparative measurements of human fetal eye structure. A digital caliper was used to measure the eye diameter and pupil size in wet tissues fixed in paraformaldehyde (PFA) and frozen sections in Tissue-Tek OCT in EtOH-exposed samples and unexposed controls. For each group, eyes and pupils were measured, and the mean diameter was used for the calculations in the table.

	Control, GA 11-14 (n = 10, ±SD)	Control, GA 14.1-21 (n = 10, ±SD)	EtOH, GA 11-14 (n = 10, ±SD)	EtOH, GA 14.1-21 (n = 10, ±SD)
Eye Diameter (wet specimen in PFA), mm	5.8 ± 0.8; *p* < 0.05	10.2 ± 1.5; *p* = 0.055 (n.s.)	5.0 ± 0.6; *p* < 0.05	9.5 ± 1.7; *p* = 0.060 (n.s.)
Eye Diameter (histological sections in OCT), mm	5.4 ± 0.9; *p* < 0.05	9.9 ± 0.8; *p* < 0.05	4.9 ± 0.7; *p* < 0.05	9.0 ± 0.9; *p* < 0.05
Pupil Diameter (wet specimen in PFA), mm		5.3 ± 0.4; *p* < 0.05		4.6 ± 0.7; *p* < 0.05

**Table 4 ijms-24-00135-t004:** Influence of EtOH exposure on eye diameter in fetuses of Caucasian and African American mothers. For each fetus, both eyes were measured, and the mean diameters were used for the group calculations in the table in EtOH-exposed samples and unexposed controls. EtOH exposure was associated with significantly reduced eye size in both racial groups during the 1st trimester.

	Race	White		African American	
		1st Trimester (mm)	2nd Trimester (mm)	1st Trimester (mm)	2nd Trimester (mm)
EtOH Exposure					
Control (n = 20)		4.1 ± 0.2	7.4 ± 0.5	3.52 ± 0.1	6.72 ± 0.4
EtOH (n = 20)		3.34 ± 0.3; *p* = 0.007	6.76 ± 0.6; *p* = 0.05	2.26 ± 0.2; *p* = 0.0005	7.42 ± 0.6; *p* = 0.05

## Data Availability

This study collected demographic, behavioral, and laboratory data from normal healthy women, and from women who drank alcohol during pregnancy. Our research team supports all these activities and has developed a data sharing plan. We also recognize that additional benefits from data sharing may arise in the future that are not apparent at this time, and we are prepared to work specifically with NIH in addressing all requests for raw data. At the present time, we have not deposited any of this raw data in an existing databank but will make the data available to other investigators on request, in a manner consistent with NIH guidelines. Consistent with NIH policy, shared data will be rendered “free of identifiers that would permit linkages to individual research participants and variables that could lead to deductive disclosure of the identity of individual subjects” (http://grants2.nih.gov/grants/policy/data_sharing, 1 September 2006). With this caveat observed, data will be made available to the NIH/NICHD/NIAAA. Sufficient identifiers will be provided to the NIH so that research participants can be assigned a Global Unique Identifier (GUID), which is a universal subject ID that protects personally identifiable information (PII). Using the GUID, NDAR can bring together multiple types of data collected from a single participant, regardless of where and when that data was collected. Biological samples (blood, serum, exosomes and RNAs) and data that are shared will be completely free of identifiers that would permit linkages to individual research participants. We will make biological samples, de-identified data, and associated documentation available to users only under a data sharing agreement that provides for: (1) a commitment to using the data only for research purposes (2) a commitment to securing the data using appropriate computer technology; and (3) a commitment to destroying or returning remaining samples after analyses are completed. Intellectual property and data generated under this project will be administered in accordance with both University and NIH policies, including the NIH Data Sharing Policy and Implementation Guidance of 5 March 2003. As the FAIR data bank receives approval from the NIH, the data will be made available to that group as well. The NIH will be implementing a new specific policy regarding data sharing https://grants.nih.gov/grants/guide/notice-files/NOT-OD-21-014.html, as of 25 January 2023. We will adopt that policy also. Darbinian and Selzer will review and approve abstracts and manuscripts reporting the results of this study. All authors will be allowed to publish additional papers using the data in relation to their individual contributions. Decisions regarding publication are necessary in order to disseminate the data in an optimal fashion to the field, to avoid duplication of publication, and to maximize the value of the project.

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
