# Peer review of "Molecular Markers in Maternal Blood Exosomes Allow Early Detection of Fetal Alcohol Spectrum Disorders"

_ijms, 2022, doi:10.3390/ijms24010135_

Round 1

Reviewer 1 Report

Manuscript by Darbinian et al., in this manuscript titled “Molecular Markers in Maternal Blood Exosomes Allow Early 2 Detection of Fetal Alcohol Spectrum Disorders” have described very nicely the effects of alcohol exposure on prenatal developmental abnormalities, including small eyes, face and brain, and neurobehavioral deficits. They have studied fetal brains and eyes at 9-22 28 weeks’ gestation and maternal blood samples for morphometry, protein, and RNA expression, and apoptotic signaling.

EtOH use by mothers was found to be associated with reduced fetal eye diameter, brain size, and markers of synaptogenesis and increased caspase-3 activity. Levels of myelin basic protein correlated highly with morphological abnormalities. EtOH exposure was related to reduced MBP. This study has shown FB-E in the mother’s blood may contain markers, particularly MBP, that predict FASD.

This is a very important study and will have a direct impact on the field provided tested in large number of samples. Paper is well written, and authors have presented and summarized the data very nicely.

Author Response

A: We appreciate Reviewer 1’s comments.  

Reviewer 2 Report

This study seeks to find early markers of fetal alcohol damage by analysing biological samples from pregnancy terminations in both exposed and unexposed pregnancies. This is a very worthy area of investigation, since early diagnosis especially before birth would allow earlier intervention, but remains elusive. The potential for using non-invasive biomarkers in maternal serum, indicative of fetal brain characteristics, is particularly promising as an area of investigation. The manuscript is well presented throughout, although the research questions/study objectives and their rationale need to be set out more clearly from the beginning, to avoid over-interpretation of secondary analyses and in particular possible post-hoc analyses. The study has a number of methodological issues, many stemming from the combination of the chosen design and the very limited sample size (although possibly typical of these type of detailed measurements). I note that I cannot comment on the novelty or merit of the experimental techniques used to measure biomarkers in tissues.

One major fault of the study lies with the design of comparing 20 alcohol-exposed ‘cases’ to 20 unexposed ‘controls’ (of which more later). As the exposure wasn’t assigned in an experimental setting, but observed as reported by women seeking the termination, and alcohol use correlates markedly with other health behaviours and socioeconomic factors which also influence the outcomes under study, this paper can’t and shouldn’t conclude that the observed differences in outcomes are solely due to prenatal alcohol exposure. Indeed, more samples would be needed to draw such conclusions, with a much higher number of participants to allow adjustment for or stratification by such other factors (eg table 1 itself shows differences between exposed and unexposed pregnancies in terms of maternal age, BMI and smoking, which on their own could at least contribute to the observed effects).

Generally there is a problem with power, especially since the authors measured many outcomes in relation to the exposed and control subjects, and reported on a number of different analyses also including stratification by trimester, sex, ethnicity and level of alcohol exposure. Such a high number of tests is not supported by the small number of independent samples measured (so the authors should either vastly expanding on their measured samples, or forego some stratified analyses). In terms of power calculation, only one outcome is cited. It should be much clearer from the beginning that this is the ‘primary analysis/outcome’, and other are secondary outcomes or follow-up research questions. It also isn’t clear how Bonferroni corrections were applied (correcting for which number of independent tests?). Moreover, the only cited power calculation assumes an implausibly large effect size (60% of samples having eye defects), whereas earlier in the Introduction the authors themselves admit to FASD being difficult to predict since most alcohol exposed pregnancies do not end up showing linked phenotypes.

Another problem I found with this paper is the presentation of figures, in which each data point on the bar charts seem to represent an individual sample (relative to a certain fetus), rather than the mean of exposed and unexposed groups being compared to each other (Fig 1). More clarity in research questions from the outset will also help with planning the analyses and presenting the results in a coherent and correct way. It is also unclear how many samples were included in these analyses (eg Fig 1), as there are 5 bars for control and 5 for exposed samples in each trimester, which make up 20 overall and not 20 exposed and 20 unexposed.

The authors should state clearly what the scope of each analysis is, as there seem to be a very large number of correlation/association analyses (everything Vs everything) and it isn’t clear to the reader what their significance is.

A minor point is that this is not, in fact, a ‘case-control’ study, since study subjects were compared according to their exposure at baseline, and not sampled according to the outcome (case= with outcome, control= without outcome). This is an important distinction in epidemiological studies, and I suggest the authors use a more appropriate terminology of ‘comparison between exposed and matched unexposed samples’ to avoid confusion.

In general, I recommend the authors seek the input of an epidemiologist/statistician skilled in population health sciences to advise on study design and statistical analyses protocol.

I also recommend thorough checking and proofreading of the paper to correct some mistakes (Table 2 before table 1? Mean value for exposed group in African Americans higher than controls in Table 3) or inaccuracies (first paragraph of Discussion: “These two findings may allow us to conduct clinical trials to determine whether these molecular markers can predict the emergence of FASD” – clinical trials aren’t suitable for this, large-scale population-based studies should be carried out instead).

Author Response

Q: The study has the very limited sample size (although possibly typical of these type of detailed measurements).

A: We agree with Reviewer 2 that a larger number of human samples for detailed measurements are desirable, but there is a trade-off between sample size and the number of variables to be controlled.  If we used all the available samples, and if we had continued to add more fetuses after the initial experiments, we would have had 153 alcohol-exposed and 71 non-exposed controls, but we could not control for all the factors, such as gestational age, fetal gender, maternal race, socioeconomic status, or use of other substances, any of which could confound the results.  Given a large enough sample, we could hope that these and other variables would cancel themselves out, but by its nature, the sample size would have to be limited and we could not afford financially to do all the biomarker determinations that we surveyed. For those reasons,  we began by selecting several molecular markers, based on the literature and our previous studies (particularly Darbinian et al, 2021), and applied a case-controlled method, matching for several variables, using the first 20 EtOH-exposed cases and 20 gestational age- and sex-matched unexposed controls in the order they were obtained, for our initial survey as described in Methods. These ideal cases all had intact brains and eyes, matched maternal blood samples for exosome isolation, and other complete parameters. Then we focused on the molecular marker (myelin basic protein, MBP) that showed the best correlation with the EtOH exposure, and included the larger numbers that had been collected by the time the experiments were initiated (n=60 EtOH-exposed and 60 gestational age-matched controls), in which anatomical preservation was sufficient to measure eye diameter, and for which matching maternal blood samples were available for exosome isolation, for the final experiments.  Thus, all the measurements and assays were studied prospectively. Naturally, it would have been unethical to design an experiment in which the exposure to alcohol was assigned prospectively.  Only in that sense was the experimental design retrospective.  Although a more definitive interpretation would require larger numbers and more controlled variables, we believe this study is large enough to justify a prospective clinical study to test the value of fetal exosomal MBP as a biomarker to predict the emergence of FAS postnatally. The purpose of the study is now better explained in Introduction and Discussion.

Q: One major fault of the study lies with the design of comparing 20 alcohol-exposed ‘cases’ to 20 unexposed ‘controls’ (of which more later). As the exposure wasn’t assigned in an experimental setting, but observed as reported by women seeking the termination, and alcohol use correlates markedly with other health behaviours and socioeconomic factors which also influence the outcomes under study, this paper can’t and shouldn’t conclude that the observed differences in outcomes are solely due to prenatal alcohol exposure. Indeed, more samples would be needed to draw such conclusions, with a much higher number of participants to allow adjustment for or stratification by such other factors (eg table 1 itself shows differences between exposed and unexposed pregnancies in terms of maternal age, BMI and smoking, which on their own could at least contribute to the observed effects).

Generally there is a problem with power, especially since the authors measured many outcomes in relation to the exposed and control subjects, and reported on a number of different analyses also including stratification by trimester, sex, ethnicity and level of alcohol exposure. Such a high number of tests is not supported by the small number of independent samples measured (so the authors should either vastly expanding on their measured samples, or forego some stratified analyses). In terms of power calculation, only one outcome is cited. It should be much clearer from the beginning that this is the ‘primary analysis/outcome’, and other are secondary outcomes or follow-up research questions. It also isn’t clear how Bonferroni corrections were applied (correcting for which number of independent tests?). Moreover, the only cited power calculation assumes an implausibly large effect size (60% of samples having eye defects), whereas earlier in the Introduction the authors themselves admit to FASD being difficult to predict since most alcohol exposed pregnancies do not end up showing linked phenotypes.

A: See above.  We did increase the number of cases to n=60, and also included data correlated with dose of EtOH exposure (Figure 4a, 4c, 4d). Bonferroni correction for multiple comparisons were applied within figures, where appropriate. As stated, power calculation assumes a large effect size (60% of samples having eye defects). Although in the Introduction we cited literature that indicates that most alcohol exposed pregnancies do not end up showing linked phenotypes, if FASD is diagnosed, eye defects are very frequent.  Indeed, what we describe in the present study suggests that many mild cases of FASD may be unreported and that in these, small eye size may be common, but not measured. This is now indicated in Discussion.

Although we did not control for all possible variables, the data presented show that 10 (out of 20) mothers were White and 10 were Black. As for Hispanics or non-Hispanics, 3 out of 20 were Hispanics (both White Hispanics or Black Hispanics), and 17 were non-Hispanics. Although we did not control for it, the prevalence of tobacco use was similar in EtOH-exposed and control mothers (4 individuals, or 25%, vs 5 individuals, or 20%). In future studies, we plan to determine the effects of tobacco use on fetal brain development. Finally, 10 out of 20 fetuses were males, and 10 females. This is now included in Methods.

Q: Another problem I found with this paper is the presentation of figures, in which each data point on the bar charts seem to represent an individual sample (relative to a certain fetus), rather than the mean of exposed and unexposed groups being compared to each other (Fig 1). More clarity in research questions from the outset will also help with planning the analyses and presenting the results in a coherent and correct way. It is also unclear how many samples were included in these analyses (eg Fig 1), as there are 5 bars for control and 5 for exposed samples in each trimester, which make up 20 overall and not 20 exposed and 20 unexposed.

A: In Figure 1 and others, 20 EtOH-exposed fetuses were studied, 10 in the 1st trimester and 10 in the 2nd trimester. The fetuses were individually matched with an unexposed control. We have changed Figure 1 to clarify the issues raised.  In the original figure, each bar represented the mean of two samples from the same fetus, each determination performed in triplicate, and the error bars incorporated the triplicate measurements, in order to show the overall variability. We now present the data in scatter plots, each dot representing one fetus.  We have eliminated the error bars, to avoid confusion.  As before, all p values reflect the number of fetuses, not the number of measurements.  The figure legend has been expanded to clarify all these points. 

Q: The authors should state clearly what the scope of each analysis is, as there seem to be a very large number of correlation/association analyses (everything Vs everything) and it isn’t clear to the reader what their significance is.

A: The meaning of the various p values is now better explained in the figure legends. The reformatting of Figure 1 simplifies this.  In Figure 1, each plate compares each alcohol-exposed fetus with each matched control. The data were analyzed by ANOVA, and, because two comparisons are made in each panel (one for 1st trimester and one for the 2nd trimester) a Bonferroni correction of two is used to multiply the p values.  We stipulate in Methods that Bonferroni corrections are made to adjust for multiple comparisons within a frame.  We did not feel it was justified to apply additional corrections across multiple frames or multiple figures.

Q: A minor point is that this is not, in fact, a ‘case-control’ study, since study subjects were compared according to their exposure at baseline, and not sampled according to the outcome (case= with outcome, control= without outcome). This is an important distinction in epidemiological studies, and I suggest the authors use a more appropriate terminology of ‘comparison between exposed and matched unexposed samples’ to avoid confusion.

A: Thank you.  In Materials and Methods, in some places, we have replaced “case-control study’ with “comparison between exposed and matched unexposed samples.” However, we do need to indicate that for many of the figures, we are matching each individual EtOH-exposed fetus with its control, and we have spelled this out where appropriate. 

Q: In general, I recommend the authors seek the input of an epidemiologist/statistician skilled in population health sciences to advise on study design and statistical analyses protocol.

A: We did consult a very experienced statistician, Dr. Huaqing Zhao, from the Department of Biomedical Education and Data Science, Lewis Katz School of Medicine at Temple University, who helped design our study and interpret the results. We did not include an epidemiologist, because we did not consider our study epidemiological at this point. In a future study, we plan to correlate the molecular marker from fetal brain exosomes with larger numbers of cases to see how well the markers predict the occurrence of FASD postnatally, and this study will involve an epidemiologist.  This is now better explained in Introduction.

In all studies, two groups were used for statistics (EtOH-exposed cases vs non-exposed Controls). In Figure 8, two statistical assays were used (Spearman’s correlation on exact Two-Tailed Probabilities, and Students t-test) and these yielded similar p-values. For Scatter plots, n’s are presented within the graph.  Numbers are given under the graph when Spearman’s correlation was used, as is usually suggested for n=5 to 20).

Q: I also recommend thorough checking and proofreading of the paper to correct some mistakes (Table 2 before table 1? Mean value for exposed group in African Americans higher than controls in Table 3) or inaccuracies (first paragraph of Discussion: “These two findings may allow us to conduct clinical trials to determine whether these molecular markers can predict the emergence of FASD” – clinical trials aren’t suitable for this, large-scale population-based studies should be carried out instead).

A: As suggested by Reviewer 2, we changed the order of Table 1 and Table 2. We pointed out that only the 1st trimester was studied for the African American group in Table 3. We also replaced “clinical trials” with “large-scale population-based studies,” as suggested.